# Biochemical Characterization of Human Salivary Extracellular Vesicles as a Valuable Source of Biomarkers

**DOI:** 10.3390/biology12020227

**Published:** 2023-01-31

**Authors:** Valentina Mangolini, Alice Gualerzi, Silvia Picciolini, Francesca Rodà, Angela Del Prete, Luana Forleo, Rudy Alexander Rossetto, Marzia Bedoni

**Affiliations:** 1IRCCS Fondazione Don Carlo Gnocchi ONLUS, 20148 Milano, Italy; 2Dipartimento di Medicina Molecolare e Traslazionale, Università degli Studi di Brescia, 25122 Brescia, Italy; 3Clinical and Experimental Medicine PhD Program, University of Modena and Reggio Emilia, 42100 Modena, Italy; 4Ordine dei Biologi Lombardia, 20090 Milano, Italy

**Keywords:** saliva, Raman spectroscopy, biomarkers, extracellular vesicles, diagnostics

## Abstract

**Simple Summary:**

Extracellular vesicles (EVs) are an emerging source of biomarkers for a plethora of human disorders, ranging from neurodegenerative to cancer disorders. Most works focusing on biomarker discovery use blood as the EV source; however, although informative, EVs from blood are not easily separated from other particles and proteins circulating in blood vessels. Other authors have proposed the use of saliva as an EV source, while also finding limitations in the downstream analysis due to the limited amount of salivary EVs. Herein, we propose the isolation of salivary EVs and their subsequent characterization by Raman spectroscopy to obtain a comprehensive biochemical characterization that can be easily translated to diagnostics. The proposed method was successfully applied. The sensitivity of the Raman characterization of EVs was not limited by the low concentration of salivary EVs compared to serum EVs, and was able to provide a comprehensive characterization of EVs in a high throughput and repeatable manner. Consequently, our data suggest a new perspective on the use of salivary EVs when the interference of lipoproteins might hinder biomarker detection or when blood withdrawal is hampered. Raman spectroscopy can represent a turning point in the application of salivary EVs in clinics.

**Abstract:**

Extracellular vesicles (EVs) are natural nanoparticles secreted under physiological and pathological conditions. Thanks to their diagnostic potential, EVs are increasingly being studied as biomarkers of a variety of diseases, including neurological disorders. To date, most studies on EV biomarkers use blood as the source, despite different disadvantages that may cause an impure isolation of the EVs. In the present article, we propose the use of saliva as a valuable source of EVs that could be studied as biomarkers in an easily accessible biofluid. Using a comparable protocol for the isolation of EVs from both liquid biopsies, salivary EVs showed greater purity in terms of co-isolates (evaluated by nanoparticle tracking analysis and Conan test). In addition, Raman spectroscopy was used for the identification of the overall biochemical composition of EVs coming from the two different biofluids. Even considering the limited amount of EVs that can be isolated from saliva, the use of Raman spectroscopy was not hampered, and it was able to provide a comprehensive characterization of EVs in a high throughput and repeatable manner. Raman spectroscopy can thus represent a turning point in the application of salivary EVs in clinics, taking advantage of the simple method of collection of the liquid biopsy and of the quick, sensitive and label-free biophotonics-based approach.

## 1. Introduction

Extracellular vesicles (EVs) are known to be natural nanoparticles released by all cells in the body under physiological and pathological conditions. EVs perform their biological functions by delivering proteins, metabolites and nucleic acids to the recipient cells, thereby ensuring cell communication. It has been shown that EVs carry and express the protein markers typical of the cell of origin, both inside the vesicles and on their membrane surface. Indeed, the main characteristic of EVs is to reflect the condition and content of the parental cell [1]. As a result, EVs have been increasingly studied over the years as biomarkers of a variety of diseases, including neurological disorders [2]. 

To date, most studies on EVs use blood as the liquid biopsy and EV source. Disadvantages related to the use of serum and plasma to isolate EVs are the presence of a high protein concentration (for example, albumin) and of lipoproteins that share dimensions and density with EVs, thus representing co-isolates in most isolation procedures [3]. Moreover, there are a plethora of pre-analytical variables that can affect the quality of the blood preparation, hamper the reproducibility of EV isolation and represent an obstacle in the validation of blood EVs as biomarkers [4]. As an example, it is not yet clear whether a fasting situation is necessary for the isolation of EVs; the plasma concentration of EVs has been measured using Nanoparticle Tracking Analysis (NTA) before and after eating a meal, and it showed that concentration did not vary significantly after eating a high-fat meal [5]. However, the plasma concentration of EVs is strongly correlated with the concentration of very low density lipoproteins (VLDL) in plasma and triglyceride in serum after meals, suggesting that the number of particles measured through NTA may be affected by the presence of these macromolecules. Furthermore, attention should also be paid to the type of test tube used. Indeed, samples obtained using tubes with Ethylenediaminetetraacetic Acid (EDTA) as blood anticoagulant showed the highest plasma concentrations of EVs, supporting the idea that EDTA promotes the formation of micro-vesicles due to platelet activation [5]. 

A biological fluid that could help in overcoming some of the limits of the use of blood as the EV source is saliva. Saliva is an optimal biological fluid for diagnostics as it contains a mixture of systemic molecules able to give information on the physiological and pathological state of a subject [6,7,8,9]. Furthermore, it presents the advantage of being a non-invasive and low-cost sampling method. Indeed, salivary samples are more easily collected compared to blood, as no specialized professionals are required [10], even though the composition of saliva can change according to the circadian cycle [11]. Due to its remarkable diagnostic potential, human saliva has been widely explored for the identification of oral [12,13] as well as systemic diseases [6,7,8,9,14,15]. Among its components, it has been reported that human saliva harbors EVs, namely salivary extracellular vesicles (SEVs) [16,17] that have a higher degree of purity compared to EVs from serum or plasma [18]. Due to the low amount of co-isolated proteins and the absence of lipoproteins, SEVs have been proposed as potential biomarkers of diseases ranging from cancer [19,20] to traumatic brain injury (TBI) [21]. Moreover, knowing that salivary glands release α-synuclein in saliva [22], SEVs have been investigated in people with Parkinson’s disease (PD) [23], suggesting the possibility of using SEVs as potential diagnostic biomarkers in multiple diseases, including neurodegenerative diseases, where early diagnosis is urgently needed. Nonetheless, the main limitation for the use of SEVs seems to be the paucity of EVs in saliva and, consequently, the limit of detection for the downstream method. 

For these reasons, in this study, we propose the use of Raman spectroscopy (RS), a biophotonics-based method, for the characterization of the EV salivary content and the comparison of the physico-chemical features of EVs isolated from both saliva and serum from the same subjects. RS allows the evaluation of the overall biochemical composition from a tiny volume of sample without labelling, providing a unique spectral fingerprint of each sample with peaks corresponding to specific molecular bond vibrations. This technique has already been applied for the analysis of blood, saliva or EVs isolated from biofluids such as plasma or serum, or from specific cells, with diagnostic purposes, demonstrating the ability of RS to identify a biochemical profile of the sample, showing simultaneously its content in proteins, lipids and nucleic acids [24,25,26,27].

In this preliminary study, we describe a straightforward procedure for the collection of salivary samples, the isolation of SEVs and the subsequent analysis by RS [6,7,8,9]. The concomitant analysis of SEVs and serum-derived EVs from the same subjects demonstrates differences that might relate to the cell source of the EV samples, to the protein corona and/or to co-isolated macromolecules. To the best of our knowledge, this is the first time that the biochemical characterization of EVs from saliva has been performed, and that RS analysis of EVs from serum and saliva were performed in parallel for the same subject, suggesting a new perspective for the use of SEVs when the interference of lipoproteins might hinder biomarker detection or when blood withdrawal is hampered.

## 2. Materials and Methods

### 2.1. Subject Recruitment and Sample Collection

In this study, five volunteers with good health status, and with no previous diagnosis of cardiovascular diseases, neurological or severe systemic disorders, were enrolled (average age 33.4 y/o; all subjects were female and non-smokers). According to the Helsinki Declaration, ethical approval was obtained from the Institution Ethical Committee and all recruited subjects signed informed consent forms before the collection of serum and saliva samples. Serum was obtained by the centrifugation of blood at 2500 g for 10 min at room temperature, and then stored at −80 °C before analysis. For the collection of saliva, Salivette^®^ swabs were purchased from Sarstedt (Sarstedt AG & CO, Numbrecht, Germany). Each Salivette^®^ contains a cotton swab that can filter saliva and retain any food residues. The saliva collection procedure was performed following the manufacturer’s instructions. Briefly, the swab was placed in the mouth and chewed for one minute, at least two hours after the last meal and teeth brushing. Then, the swab was centrifuged at 1000× *g* for 5 min in order to recover the biofluid. Saliva samples were stored at −20 °C before analysis. This method was previously reported for the collection of saliva for Raman analysis, demonstrating no interference in the subsequent spectroscopic evaluation [6,7,8,9]. 

### 2.2. Isolation of Extracellular Vesicles

EVs were isolated from the serum and saliva of the enrolled subjects by Size Exclusion Chromatography (SEC) (qEV/70 nm series; Izon Science, Christchurch, New Zealand). Prior to loading onto the SEC column, serum and saliva samples were thawed and centrifuged at 10,000× *g* for 10 min. Saliva samples were also concentrated using Amicon 3K filters (Amicon Ultra, Sigma-Aldrich, St. Louis, MO, USA). A total of 500 μL of each serum or saliva sample was loaded in the SEC column and fractions (500 μL each) from 4 to 16 were collected. 

### 2.3. EV Size Distribution

SEC fractions from 7 to 11 obtained from both serum and saliva were pooled together and characterized by NTA (NanoSight NS300; Malvern, Panalytical LTD, Malvern, UK) to evaluate the size distribution and concentrations of EVs. For the analysis, all samples were diluted in fresh filtered phosphate buffered saline (PBS) (saliva 1:10–serum 1:100) and then injected in the sample chamber through a syringe pump that provides a continuous flow of new particles into the sample chamber. Recordings of the movements of particles were collected for 60 s, five times for each sample. In parallel, the total protein content of the EV-containing fractions (7–11) was measured by BCA assay (Pierce BCA assay kit, Scientific, Waltham, MA, USA) after EV lysis in 1X RIPA buffer. 

### 2.4. EVs Purity Assessment

In order to determine the purity of both serum EVs and SEVs, the COlorimetric NANoplasmonic (CONAN) Method was performed [28,29]. To start the CONAN assay, spherical gold nanoparticles (GNPs) were prepared using the classic Turkevich’s citrate reduction method [30]. Briefly, 20 mL of tetrachloroauric acid solution (HAuCl_4_) 1 mM was prepared and boiled under stirring. Then, 2 mL of 1% sodium citrate was added and left to stir for 15 min. Thereafter, the solution was placed in an ice and H_2_O bath. Finally, the diameter and concentration values of GNPs were calculated from the UV spectrum values by referring to the tables in the literature [31]. GNPs with a concentration of 6nM were used for the CONAN assay, mixing EV samples with GNPs in a ratio of 1:1. The aggregation index (AI) % for each sample was calculated in order to assess the purity of the EVs. GNPs in H_2_O and GNPs in PBS were used as controls. For EV samples, the preparation can be considered as having an acceptable degree of purity when the AI is not greater than 50% [28].

### 2.5. Raman Analysis

Before the Raman analysis, the EV-containing fractions (7–11) obtained by SEC were ultra centrifuged at 100,000× *g* for 70 min (L7-65; Rotor SW60; Beckman Coulter, Brea, CA, USA). A drop of freshly isolated and concentrated EVs (4 µL of serum EVs and 20–30 µL of SEVs) was deposited on a CaF_2_ disk and dried at room temperature. The Raman analysis was performed using a Raman microscope Aramis (Horiba Jobin-Yvon, Palaiseau, France), equipped with a 532 nm laser source, and following a previously described method for the bulk characterization of EVs [32,33,34]. First, the instrument was calibrated using the silicon reference band at 520.7 cm^−1^. For all the analysis, a 50× objective (Olympus, Tokyo, Japan) was used, with acquisition time of 30 s for two accumulations, diffraction grating at 1800 grooves/mm, 400 µm entrance slit and hole at 600 µm. Spectra were acquired using LabSpec6 software (Horiba Scientific) in the spectral ranges 500–1800 cm^−1^ and 2700–3200 cm^−1^ with a spectral resolution of 0.8 cm^−1^/step for the first range and 0.6 cm^−1^/step for the second range. For all samples, about 20 spectra were acquired for each drop, both at the edges and in the center of the drop area. Afterwards, all the acquired spectra were fit with a polynomial baseline, resized on the reference band at 1554 cm^−1^ for the first range and 2938 cm^−1^ for the second range. Normalization was performed through unit vector normalization in order to compensate for autofluorescence and background interference. Finally, adapting the Mihàly’s protocol for IR spectroscopy [35], Raman spectra were used to measure the spectroscopic ratios of nucleic acids–proteins (AC/P) and proteins–lipids (P/L) for both serum EVs and SEVs, as previously described [33].

### 2.6. Statistical Analysis 

Raman spectra were analyzed by descriptive and multivariate statistical analysis using OriginPro2022 (OriginLab, Northampton, MA, USA) as previously described [32,33]. Average Raman spectra were obtained for serum-derived EVs and SEVs. The Mann–Whitney non-parametric test was used to compare data obtained from EV characterization (i.e., NTA analysis). The index was identified as statistically relevant for *p*-values < 0.05.

## 3. Results

### 3.1. EV Physico-Chemical Characterization 

The EVs were isolated from all considered serum and saliva samples by SEC. The NTA analysis was performed on the pooled SEC fractions (from 7 to 11) containing EVs to evaluate their dimensions and concentrations. As expected, serum EVs (mean concentration: 2.4 × 10^10^ particles/mL) proved to be more concentrated than SEVs (mean concentration: 3.2 × 10^9^ particles/mL) (*p* = 0.012, Mann–Whitney test; Figure 1a). On the other hand, serum EVs and SEVs showed comparable size distribution (Figure 1b) and approximately the same mean sizes at 136 nm and 135 nm, respectively. Considering the total amount of proteins measured by BCA, EV samples isolated from serum contained a higher concentration of proteins compared to SEVs (Figure 1c) (*p* = 0.019, Mann–Whitney test), while the amount of proteins per particle was increased in SEVs compared to serum EVs, with a statistically significant difference (*p*= 0.036, Mann–Whitney test) (Figure 1d). 

To investigate the source of the increased protein concentration per particle, the CONAN assay [28] was performed to evaluate the degree of purity of serum EVs and SEVs and to verify whether the protein corona could account for the increased protein concentration. As shown in Figure 1e, GNPs in contact with SEVs and serum EVs changed their localized surface plasmon resonance (LSPR) peak indicating aggregation. The CONAN results showed a mean AI equal to 14.33% for the SEV preparation, while serum EVs presented an AI equal to 26.4%, indicating a higher purity for the SEV preparation compared to the serum-derived EVs (Figure 1f).

### 3.2. Raman Analysis

The Raman analysis was performed starting from previously optimized parameters for EV characterization [32,33,34]. The biochemical fingerprint of EVs was obtained investigating two spectral ranges: 500–1800 cm^−1^ and 2700–3200 cm^−1^. For the EVs isolated from serum (n = 5) and saliva (n = 5) samples, the average spectra (±standard deviation) were calculated demonstrating (i) a good signal-to-noise ratio in both samples and (ii) the applicability of the proposed spectroscopic procedure for saliva-derived EVs (Figure 2a,b). The fingerprint of EVs from both biofluids showed peaks in the Amide I and Amide III regions as well as bands in the spectral ranges accounting for lipid components and nucleic acids, as expected from previous studies [32,36,37]. For an easier visualization of the major spectral differences, the average spectra of serum-derived EVs and SEVs are overlapped in Figure 2c. As shown, the differences between the experimental groups are more prominent in the first spectral range (500–1800 cm^−1^). Indeed, some peaks overlap and others are shifted, in particular the average spectrum of SEVs presents a shifted peak at 971 cm^−1^, relating to the C-C stretch of carbohydrates. 

Besides the presence of differences in the relative intensities of some peaks, serum EVs and SEVs presented characteristic peaks in their average spectra, as indicated by arrows in Figure 2c. As for SEVs, in the nucleic acids region, peaks at 618 cm^−1^ and 787 cm^−1^ are present that are absent in the mean spectrum of serum EVs, as well as peaks related to carbohydrates (898 cm^−1^) and proteins (1209 cm^−1^). Furthermore, the peak at 1380 cm^−1^ relating to C-H bending is well evident in the SEV spectrum, but not in the one for serum EVs. As regards the spectral range attributable to Amide I, in the SEV spectrum there is a peak at 1616 cm^−1^, while a peak at 1666 cm^−1^ is present for serum EVs. In the second range (2700–3200 cm^−1^), the main peaks related to lipids were observed both in the serum and saliva EV spectra, with a higher intensity in the spectrum of the serum EVs. The peak assignments reported in Table 1 were based on previous literature [32,38,39].

To better elucidate the differences between the serum EVs and SEVs, the spectroscopic ratios NA/P and P/L were calculated. The NA/P and P/L were obtained by dividing the relative intensities of the nucleic acids band (720–800 cm^−1^) by the Amide I band (1600–1690 cm^−1^), and the Amide I band (1600–1690 cm^−1^) by lipids band (2750–3040 cm^−1^), respectively. As shown in Figure 3a, the spectroscopic NA/P ratio obtained from SEV was statistically higher than the ratio obtained from serum EVs (*p* = 1.7 × 10^−7^—Mann–Whitney test) suggesting a difference in the nucleic acids load between the two experimental groups of samples. On the other hand, the spectroscopic L/P ratio showed no significant differences (Figure 3b, *p* > 0.05—Mann–Whitney test). 

## 4. Discussion

In recent years, EVs have become an important object of study as potential disease biomarkers. However, one of the main difficulties is related to their small size, which makes it difficult to efficiently separate and prepare EVs, especially from complex liquid biopsies such as blood [4,5,19]. To date, there are still few studies that have used saliva as the biofluid for the isolation of EVs. It was previously reported that SEVs can carry cancer-cell-specific proteins and enter saliva from the blood in patients with lung cancer [19], and brain-derived EVs (L1CAM+) have also been found in the saliva of people with Parkinson’s disease [23], as were amyloid-beta and p-tau markers of Alzheimer’s disease [40]. However, one of the major limits for the use of saliva as a source of EVs for diagnostic purposes is the common knowledge that saliva EV concentration is significantly lower than in blood [41], although it contains most of the major soluble constituents of blood.

In the present preliminary study, we proposed the comparison of EVs isolated from serum and from saliva, in order to deepen the understanding of their biochemical features and reconsider the possibility of using SEVs as potential biomarkers in diagnostics. First of all, the collection of the two biological fluids took place, respectively, through blood sampling and through the mastication of salivary swabs (Salivette^®^). The use of a cotton swab for saliva collection allows the removal of any food residues and the filtering of any viscous salivary component in the saliva, which would be obstacles for the isolation of EVs [19]. We herein report that serum EVs are more concentrated than SEVs, confirming the data in the literature [41], although the concentration of serum EVs observed by NTA could be overestimated due to the co-isolation of lipoproteins, especially very low density lipoproteins (VLDL) that are smaller in size than EVs but have the same density. On the other hand, saliva does not contain lipoproteins, thus resulting in a lower concentration of isolated particles in saliva. In addition, we registered a higher concentration of total protein content in serum EVs compared to SEVs, allowing us to hypothesize that either serum EVs are loaded with a consistently higher amount of proteins, or that protein co-isolation occurs, or both. Indeed, it is well known that EVs that circulate in blood have the so-called protein corona, which consists of proteins that can be found anchored to the vesicles and cannot be separated from the outer membrane of EVs during isolation, some of them being part of the protein cargo adsorbed to EVs, but also partly recruited in body fluids after vesicle shedding [42,43]. The hypothesis of a more consistent protein corona on serum EVs compared to SEVs was supported by the CONAN assay results that showed a higher level of purity for SEVs than for serum EVs, i.e., a reduced amount of co-isolated proteins. The principle of the CONAN assay is based on the high surface energy of the GNPs. Indeed, when the GNPs reacted with SEVs, the color of the solution immediately changed from red to blue, indicating an aggregation with the membrane proteins of the EVs that would not happen if other contaminants/co-isolates were present in the sample. Considering a 100% AI for a solution of not aggregated GNPs (GNPs in water), a lower AI% referred to a sample with few contaminants and we found that the serum EV samples were richer in co-isolates than the SEV samples. Finally, it is worth noting that the calculated ratio µg of protein per particle was greater for SEVs than for serum EVs, suggesting that SEVs could have a limited protein corona but a major protein load in their core. 

To evaluate the biochemical features of EVs from blood and saliva, we took advantage of RS, a vibrational spectroscopy method that, in recent years, has been proven to ensure high chemical specificity and a fast analytical procedure also in the EV field. Specifically, RS allowed us to rapidly and accurately characterize EVs, providing detailed information on the average composition of external and inner molecules, demonstrating its remarkable potential for both basic research and translational EV analyses. In the present study, we applied a previously optimized acquisition protocol that was already proven to be able to comprehensively evaluate the biochemistry of a sample starting from tiny volumes of EV suspension [32,33,34]. The RS analysis revealed biochemical differences between the serum EVs and SEVs, both in terms of variations in the intensity and in the presence of some characteristic peaks. Indeed, the differences in the biochemical composition found in the Raman spectra of SEV and serum EVs could explain differences in the cellular origin of EVs circulating in the two biofluids and also differences in the molecules loaded and/or co-isolated with EVs. Major differences were found in the Amide regions, thus accounting for differences both in the internal cargo and in the protein corona, and in the carbohydrate components, that could be explained by differences in the glycosylation of proteins and lipids present within vesicles. Concerning the latter observation, it has to be noted that most EVs in saliva are expected to be originated from the respiratory tract, and thus the observed differences in the carbohydrates could account for the presence of mucins within or adsorbed to SEVs. Mucin is a highly glycosylated protein that is produced by the cells of ciliated epithelium of upper airways and it is known that vesicles originated from these cells express different forms of mucin on their surface (i.e., MUC1, MUC4 and MUC16) that protect against pathogens and maintain the structure of vesicles [44]. For this reason, we can speculate that mucin might explain some of the major differences in the Raman spectra of SEVs and serum EVs, although further studies are needed to verify its presence and the cellular origin of SEVs. Furthermore, it was observed that the spectroscopic ratio NA/P was statistically different between the mean spectra of SEV and serum EVs, while the P/L ratio was comparable in the two sample preparations. In this regard, we may speculate that SEVs may have a higher cargo of nucleic acids than serum EVs, being potentially valuable cargo also for RNA biomarkers. This is in line with previous literature demonstrating that SEVs have been found to carry tRNA-derived small RNAs (tsRNAs) which represent important biomarkers for the diagnosis and prognosis of esophageal squamous cell carcinoma [20], as well as carriers of genetic biomarkers for traumatic brain injury (TBI) [21]. 

## 5. Conclusions

In conclusion, we propose the use of RS for the characterization of salivary EVs, even with diagnostic purposes. Using a comparable protocol for both liquid biopsies, SEVs showed greater purity in terms of co-isolates, including both lipoproteins and protein corona. Moreover, the sensitivity of the Raman bulk characterization of EVs is not limited by the low concentration of SEVs compared to serum EVs, and it is able to provide a comprehensive characterization of EVs in a high throughput and repeatable manner. Consequently, our preliminary data suggest a new perspective on the use of SEVs when the interference of lipoproteins might hinder biomarker detection or when blood withdrawal is hampered. Raman spectroscopy can represent a turning point in the application of salivary EVs in clinics, taking advantage of the simple method of collection of the liquid biopsy and of this quick, sensitive and label-free biophotonics-based approach.

## Figures and Tables

**Figure 1 biology-12-00227-f001:**
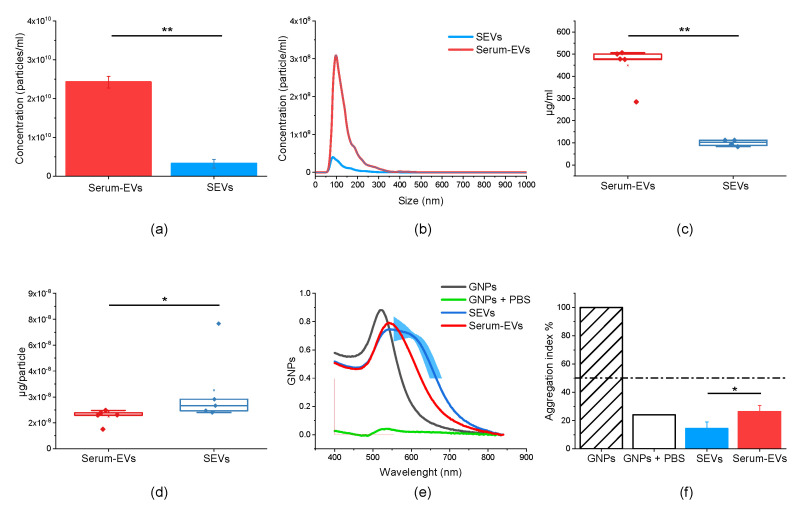
Physico-chemical characterization of extracellular vesicles isolated from serum (serum EVs, red) and saliva (SEVs, blue). (**a**,**b**) Data obtained by NTA analysis allow the comparison of particle concentration (**a**) and size distribution (**b**) of vesicles obtained from serum and saliva. (**c**) Box plot showing the values of protein concentration obtained by colorimetric microBCA protein assay. (**d**) Purity of EV preparation expressed as the ratio of μg of protein per particle, calculated by colorimetric microBCA protein assay and NTA, respectively. (**e**) UV spectra of aqueous dispersion of GNPs 6nM (black), GNPs dispersed in 10 mM PBS (green), GNPs with SEVs (blue; mean and standard deviation) and GNPs with serum EVs (red; mean and standard deviation). (**f**) Aggregation index (AI%) for each sample obtained by the nanoplasmonic assay. * indicates *p* < 0.05, ** *p* < 0.01 after Mann–Whitney test.

**Figure 2 biology-12-00227-f002:**
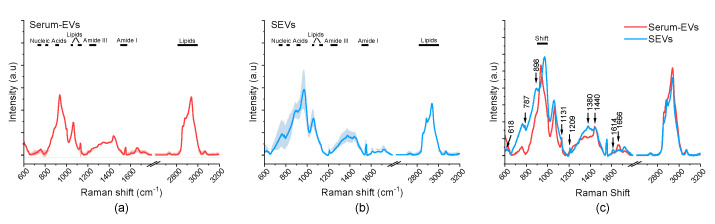
Raman fingerprint of serum- and saliva-derived EVs obtained using 532 nm laser line. (**a**) Average Raman spectrum of EVs isolated from serum samples of human healthy volunteers. (**b**) Average Raman spectrum of SEVs isolated from saliva samples of human healthy volunteers. In (**a**) and (**b**) shadow represent the standard deviation. (**c**) Overlapped spectra obtained from EVs isolated from serum (red) and saliva (blue) from the same subjects. Black lines on top of the graphs indicate some of the main macromolecular assignments.

**Figure 3 biology-12-00227-f003:**
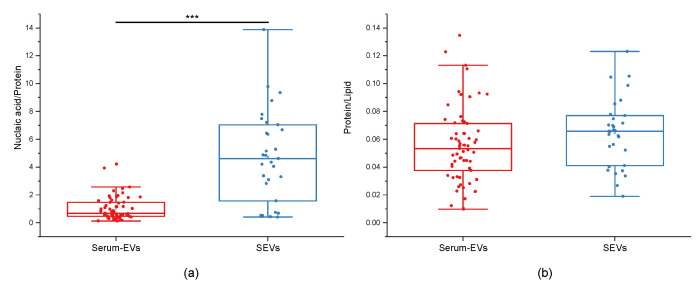
Spectroscopic nucleic acid-to-protein and protein-to-lipid ratios. Box plots showing the spectroscopic nucleic acid-to-protein ratio (nucleic acid/protein) in (**a**) and protein-to-lipid ratio (protein/lipid) in (**b**). Differences in the nucleic acid/protein and protein/lipid values obtained for serum-derived EVs and salivary EVs (SEVs) were compared. *** indicates *p* < 0.001 after Mann–Whitney test.

**Table 1 biology-12-00227-t001:** Raman peak assignments. The table reports the assignments of the main peaks that can be identified in the overlapped spectra of serum-derived EVs and SEVs.

Serum		Saliva	
cm^−1^	Assignment	cm^−1^	Assignment
760	Tryptophan	618	C-C twisting (protein)
937	C-C stretch	760	Tryptophan
1002	Phenylalanine	787	Phosphatidylserine
1064	Lipids	898	Monosaccharides
1131	Fatty acid	971	C-C stretch
1207	Hydroxyproline, Tyrosine	1060	Lipids
1440	CH_2_ lipids	1209	Tryptophan and phenylalanine mode
1600–1800	Amide I	1290–1400	CH-bending
2700–3500	Stretching vibrations of CH, NH, OH	2700–3500	Stretching vibrations of CH, NH, OH
2800–3050	Contributions from acyl chains	2800–3050	Contributions from acyl chains
2929–2940	CH_2_ asymmetric stretch	2929–2940	CH_2_ asymmetric stretch

## Data Availability

All data generated or analyzed during this study are included in this published article.

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
