# Peer review of "Biochemical Characterization of Human Salivary Extracellular Vesicles as a Valuable Source of Biomarkers"

_biology, 2023, doi:10.3390/biology12020227_

Round 1
Reviewer 1 Report
The manuscript entitled “Biochemical characterization of human salivary extracellular vesicles as a valuable source of biomarkers” by Mangolini and collaborators deals with the exploitation of Raman microspectroscopy to assess the biochemical characterization of EVs for diagnostics purposes.
In my opinion, the manuscript successfully addresses the question proposed by the authors, by an approach that seems correct and reliable.
Both the simple summary and the abstract adequately present a minimum background and the rationale of the manuscript. The results are clear and well-discussed.
Throughout all the manuscript, English is correct, and sentences are clear, even for a non-expert reader.
I suggest some minor improvements that should be applied to the manuscript before acceptance:
1. The number of volunteers enrolled in the study is quite low. Are the authors sure (by a statistical approach) that the results they obtained are significant and reliable?
2. A deeper focus should be dedicated to RS application: my suggestion for the authors is to better pinpoint the advantages of the application of RS in this field, and to clarify how useful their results could be.
Reviewer 2 Report
Dear V. Mangolini, I find your study very interesting with a very sound methodology and interesting findings. I can see your group has made quite an impact in the field of extracellular vesicles and I am delighted to see such an attempt for novel diagnostic methods.
When evaluating the study, I cannot find any major flaws. I might ask, what is the effect of a small sample size? In such experiments the participants numbers are usually small, but could this lead to some bias?
Additionally, for such a study where participants samples were collected and analysed, some ethical approval is usually required. Please include some reference.
Otherwise, congratulations on the study.
